# Tetrapeptide Ac-HAEE-NH_2_ Protects α4β2 nAChR from Inhibition by Aβ

**DOI:** 10.3390/ijms21176272

**Published:** 2020-08-29

**Authors:** Evgeny P. Barykin, Aleksandra I. Garifulina, Anna P. Tolstova, Anastasia A. Anashkina, Alexei A. Adzhubei, Yuri V. Mezentsev, Irina V. Shelukhina, Sergey A. Kozin, Victor I. Tsetlin, Alexander A. Makarov

**Affiliations:** 1Engelhardt Institute of Molecular Biology, Russian Academy of Sciences, Vavilov St. 32, 119991 Moscow, Russia; epbarykin@eimb.ru (E.P.B.); tolstova.anna.pavlovna@gmail.com (A.P.T.); anastasya.anashkina@gmail.com (A.A.A.); alexei.adzhubei@eimb.ru (A.A.A.); kozinsa@gmail.com (S.A.K.); 2Shemyakin-Ovchinnikov Institute of Bioorganic Chemistry, Russian Academy of Sciences, Miklukho-Maklaya Street, 16/10, 117997 Moscow, Russia; garifulinaai@gmail.com (A.I.G.); shelukhina.iv@yandex.ru (I.V.S.); vits@ibch.ru (V.I.T.); 3Department of Pharmacology and Toxicology, University of Vienna, Althanstraße 14 (UZA II), 1090 Vienna, Austria; 4Orekhovich Institute of Biomedical Chemistry, Pogodinskaya street 10/8, 119121 Moscow, Russia; yuri.mezentsev@ibmc.msk.ru

**Keywords:** Alzheimer’s disease, nicotinic acetylcholine receptor, cholinergic deficit, peptide drugs, molecular modeling, β-amyloid

## Abstract

The cholinergic deficit in Alzheimer’s disease (AD) may arise from selective loss of cholinergic neurons caused by the binding of Aβ peptide to nicotinic acetylcholine receptors (nAChRs). Thus, compounds preventing such an interaction are needed to address the cholinergic dysfunction. Recent findings suggest that the ^11^EVHH^14^ site in Aβ peptide mediates its interaction with α4β2 nAChR. This site contains several charged amino acid residues, hence we hypothesized that the formation of Aβ-α4β2 nAChR complex is based on the interaction of ^11^EVHH^14^ with its charge-complementary counterpart in α4β2 nAChR. Indeed, we discovered a ^35^HAEE^38^ site in α4β2 nAChR, which is charge-complementary to ^11^EVHH^14^, and molecular modeling showed that a stable Aβ_42_-α4β2 nAChR complex could be formed via the ^11^EVHH^14^:^35^HAEE^38^ interface. Using surface plasmon resonance and bioinformatics approaches, we further showed that a corresponding tetrapeptide Ac-HAEE-NH_2_ can bind to Aβ via ^11^EVHH^14^ site. Finally, using two-electrode voltage clamp in *Xenopus laevis* oocytes, we showed that Ac-HAEE-NH_2_ tetrapeptide completely abolishes the Aβ_42_-induced inhibition of α4β2 nAChR. Thus, we suggest that ^35^HAEE^38^ is a potential binding site for Aβ on α4β2 nAChR and Ac-HAEE-NH_2_ tetrapeptide corresponding to this site is a potential therapeutic for the treatment of α4β2 nAChR-dependent cholinergic dysfunction in AD.

## 1. Introduction

Alzheimer’s disease (AD) is the most common neurodegenerative disorder with over 50 million of patients worldwide [1]. Since the approval of memantine by Food and Drug Administration in 2003, no new therapeutics were developed for AD, and no disease-modifying treatments are available [2]. Currently, new therapeutic avenues are being developed on the basis of uncovering the molecular foundations of AD pathogenesis [2]. For a long period, the concepts of AD molecular pathology were focused on the role of amyloid plaques; however, it is becoming clear that neurotoxic oligomers of β-amyloid (Aβ) should be targeted as well [1,3,4,5]. Soluble neurotoxic Aβ species interact with different targets, resulting in a systemic impairment of neuronal and glial function [4,6,7]. Important targets of Aβ are brain nicotinic acetylcholine receptors (nAChRs). α4β2 and α7 nAChRs are the most abundant types of nAChRs that regulate memory, sleep, pain and cognitive processes [8,9,10]. Their activation triggers intracellular signaling, including survival-related pathways, whereas their dysfunction leads to synaptic impairment and neuronal death [11,12]. Existing data suggest that the interaction of Aβ with α4β2 and α7 nAChRs leads to selective loss of cholinergic neurons and cholinergic deficit, which is a hallmark of AD [13]. In mild AD, the region-specific loss of α4β2 nAChR correlates with the impairment of distinct cognitive domains [14]. Thus, compounds that prevent the interaction of Aβ with nAChRs could reduce neuronal loss and cognitive decline in AD. To develop such targeted compounds, we need extensive knowledge about the structure and function of Aβ-nAChR complexes and their interaction interfaces.

The ^11^EVHH^14^ region is a promising pharmacological target in Aβ, governing its zinc-dependent aggregation and cerebral amyloidogenesis in model animals [15,16]. It was recently found that ^11^EVHH^14^ site is also important for Aβ binding to α4β2 and α7 nAChRs [17]. This site contains 3 charged amino acid residues, so we hypothesized that Aβ-nAChRs interaction is mediated by the pairing of ^11^EVHH^14^ with its charge-complementary partners in nAChRs.

Here, we found that the ^35^HAEE^38^ site, which is charge complementary to ^11^EVHH^14^, is present in the α4 subunit of α4β2 nAChR. Using molecular modeling, we showed that Aβ interaction with α4β2 nAChR could occur via (Aβ) ^11^EVHH^14^:^35^HAEE^38^ (α4) interface. On the basis of this finding, we suggested that Ac-HAEE-NH_2_ tetrapeptide would (1) bind to Aβ and (2) prevent the interaction of Aβ with α4β2 nAChR, which was confirmed using surface plasmon resonance, bioinformatics approaches and electrophysiological studies.

## 2. Results

### 2.1. The HAEE Site Is Present in an Extracellular α-helix of α4β2 nAChR

It was previously shown that ^11^EVHH^14^ site in Aβ is important for interaction with α7 and α4β2 nAChRs [17,18]. ^11^EVHH^14^ motif contains several charged residues, so we hypothesized that it interacts with the other charged motif in α7 or α4β2 nAChRs on the basis of charge complementarity. To find the charged counterparts for ^11^EVHH^14^ in α7 or α4β2 nAChRs, we used the ScanProsite tool [19] (See Methods). Two such motifs were detected in α4 nAChR subunit, ^35^HAEE^38^ and ^579^KAED^582^, of which KAED is in the cytoplasmic domain, and HAEE is located in the extracellular part of α4 subunit. Hence, we assumed that the interaction between α4β2 nAChR and Aβ may be mediated by the ^35^HAEE^38^:^11^EVHH^14^ interface.

### 2.2. Aβ_42_ Can Form a Stable Complex with α4β2 nAChR through ^11^EVHH^14^:^35^HAEE^38^ Interface

Using molecular modelling, we tested the possibility of α4β2 nAChR and Aβ interaction via the ^35^HAEE^38^:^11^EVHH^14^ interface. The ^35^HAEE^38^ motif is located in the N-terminal alpha-helix of the α-subunit of α4β2 nAChR, which forms an exposed site (Figure 1A). We performed the modelling of the α4β2 nAChR structure (see Methods) and its extracellular domain was used for docking.

At Step one, to model the interaction through Aβ_42_
^11^EVHH^14^:^35^HAEE^38^ α4β2 nAChR interface, the full Aβ_42_ model was docked by targeted global docking with α4β2 nAChR where the ^11^EVHH^14^ and ^35^HAEE^38^ were indicated as potential interaction sites. The resulting dataset of 46 structures was analyzed with the in-house QASDOM server [20]. Overall, 8 structures were selected, mostly with the parallel orientation of the relevant sites, and the 3 best fitting docking models were submitted to MD simulations for 20 ns of the production run.

Step two involved refining the ^11^EVHH^14^:^35^HAEE^38^ interaction interface. Since ^35^HAEE^38^ is located in the α-helix, only three residues are exposed (Glu37 is inaccessible for binding) and only two of them can be involved in an interaction concurrently. We focused on the models where His-Glu contacts were present. To fine-tune the interaction interface centered on the Aβ_42_
^11^EVHH^14^ and α4β2 nAChR ^35^HAEE^38^ sites, we took an energy-minimized structure of the ^10^YEVHHQ^15^ fragment from Aβ_42_ and ran local docking using AutoDock Vina with different sizes of a grid box. From the dataset of docking results, a subset of structures was selected where several H-bonds were formed between the ^10^YEVHHQ^15^ and ^35^HAEE^38^ primarily through His-Glu interaction. In these structures the following combinations of the contacting residues were found: (Aβ) Glu11-His35(α4), (Aβ) His13-Glu38(α4) and (Aβ) His14-Glu38(α4). Several structures with “parallel” and “antiparallel” positioning of the Aβ_42_
^10^YEVHHQ^15^ fragment and the ^35^HAEE^38^ site were selected for the next steps of the interface modelling.

At Step three, the best fitting resulting structure of Aβ_42_ (from the Aβ_42_-α4β2 nAChR complex model) obtained in step one, was refined with Rosetta local docking server and relaxed with MD. Then it was superposed with the ^10^YEVHHQ^15^ fragment which was docked to the α4β2 nAChR ^35^HAEE^38^ site at Step two. The ^11^EVHH^14^ segment in Aβ_42_ was substituted with EVHH of the YEVHHQ peptide of the YEVHHQ-α4β2 nAChR complex structure. The resulting structure was fine-tuned by energy minimization and local docking with the Rosetta server and equilibrated by MD. At this stage, a model of the complex was created where Aβ_42_ and α4β2 nAChR were bound via the EVHH-HAEE sites (Figure 1A). The Step three modelling approach was repeated for four versions of the EVHH-HAEE interaction models and in the two cases H-bonds formed between Aβ_42_ and α4β2 nAChR by (Aβ) Glu11-His35(α4) and (Aβ) His13-Glu38(α4) in the interaction interface remained stable through the whole 100 ns of MD simulation (Figure 1B). PDB files for these structures can be found in Appendix A (structure1.pdb-structure4.pdb). Notably, “antiparallel” variants of the Aβ_42_ orientation along the α4β2 nAChR α-subunit α-helix were more stable than the “parallel” ones.

### 2.3. Ac-HAEE-NH_2_ Is Targeting ^11^EVHH^14^ in Aβ_42_

The modelling results demonstrated that a stable (parallel or anti-parallel) interaction between Aβ_42_ and α4β2 nAChR can occur via the predicted ^11^EVHH^14^:^35^HAEE^38^ interface. Hence, we assumed that a peptide corresponding to the ^35^HAEE^38^ site will bind to ^11^EVHH^14^ in Aβ and could be used to prevent the interaction of Aβ with α4β2 nAChR or to competitively displace Aβ from the complex with the receptor. We used the Ac-HAEE-NH_2_ peptide, N-acetylated and C-amidated for increased resistance to proteolytic degradation, as such Aβ-binding compound.

First, to determine the likelihood of Ac-HAEE-NH_2_ interaction with Aβ and identify the possible binding sites we performed full blind global docking of Ac-HAEE-NH_2_ to Aβ_42_., The results showed a major cluster of interactions at the ^11^EVHH^14^ site with a leading contribution of Glu11, and a less prominent cluster ^4^FRHD^7^ (Appendix A). When ^11^EVHH^14^ was indicated as a preferable interaction site in global docking (targeted docking), or a local docking using Autodock Vina was performed, the results were slightly different, with the major part of interactions centering on the His13 and Val12 residues of ^11^EVHH^14^ (Appendix A). From the targeted docking dataset, we selected 8 models in which strong hydrogen bonds between HAEE and ^11^EVHH^14^ were identified. In the majority of these structures, His14 from Aβ_42_, and Glu and His residues at the HAEE termini participated in the interactions (Appendix A). Thus, the distribution of atomic contacts in the docking dataset for the Aβ_42_ sequence identified ^11^EVHH^14^ as a preferable site for Ac-HAEE-NH_2_ binding (Figure 2A), and the targeted docking revealed possible structures of Aβ_42_-HAEE interfaces stabilized by His-Glu H-bonds (Figure 2B,C).

### 2.4. Ac-HAEE-NH_2_ Tetrapeptide Binds to Aβ_16_ In Vitro

In all mammalians, the Aβ N-terminal part 1–16 (Aβ_16_) constitutes the metal-binding domain [21,22] with a stable and well-defined conformation [23,24,25]. The domain 1–16 acts both as an autonomous molecule [26] and as an independent structural and functional unit within Aβ species of length 39–42 [27]. We have shown earlier that fragment 1–16 of Aβ (Aβ_16_) represents an adequate model for in vitro studies of the interactions that are mediated by the ^11^EVHH^14^ site [28,29,30]. Hence, we used Aβ_16_ to test the rationally predicted ability of Ac-HAEE-NH_2_ tetrapeptide to interact with Aβ in a direct binding experiment with surface plasmon resonance technology.

We found that injection of Ac-HAEE-NH_2_ over a surface with immobilized Aβ_16_ results in a dose-dependent response (Figure 3), indicating a direct peptide binding, and the calculated dissociation constant Kd was 9 ± 3 × 10^−5^ M (k_on_ = 0.37 M^−1^ s^−1^, k_off_ = 0.04 × 10^−3^ s^−1^). For the concentrations of Ac-HAEE-NH_2_ below 1 mM, the signal was insignificantly different from the reference and thus the results are not shown. In addition, 23 other tetrapeptides with a predicted charge complementarity for the ^11^EVHH^14^ region were tested in this SPR assay (Appendix A). Generally, we can conclude that the peptides designed to interact with ^11^EVHH^14^ in a parallel orientation showed better binding properties than the peptides designed to interact in an anti-parallel way. Of all the peptides tested, Ac-HAEE-NH_2_ (Kd 9 ± 3 × 10^−5^ M) and Ac-RADD-NH_2_ (Kd 1.3 ± 3 × 10^−5^ M) demonstrated the strongest binding to the Aβ_16_ (Appendix A).

### 2.5. In Silico Model of Ac-HAEE-NH_2_ Binding Interface with ^11^EVHH^14^ in Aβ_16_

To further model the HAEE-EVHH interaction in Aβ_16_ we used models 1 and 7 from the PDB:1ZE7 solution NMR structure. As for Aβ_42_, we performed global targeted docking with preferable target site specification (^11^EVHH^14^) and local docking with AutoDock Vina. Results for the global targeted and local docking are shown in Figure 4 and Appendix A.

Aβ_16_ is flexible and can adopt different conformations in solution, some of which are preferable for Ac-HAEE-NH_2_ binding. We identified a range of structures with hydrogen bonds between the Ac-HAEE-NH_2_ and ^11^EVHH^14^ regions of Aβ_16_. Of these, 22 structures were selected for further analysis with at least three hydrogen bonds between three different side-chain atoms of Ac-HAEE-NH_2_ and ^11^EVHH^14^. As shown in Appendix A, interactions mainly occur via His and Glu residues. The Ac-HAEE-NH_2_ structures in the complexes were oriented crosswise respectively to the ^11^EVHH^14^ region (Figure 4A,B) but there were some structures with a parallel orientation where three hydrogen bonds are formed between histidine and glutamic acid residues. Such structures were close to the proposed interface based on complementarity between His and Glu residues, and the interface remained stable after energy minimization in water with an AMBER99SB-ILDN force field (Figure 4C).

Since modelling results showed the presence of H-bonds between (Aβ) Glu11-His35(α4), (Aβ) His13-Glu38(α4) and (Aβ) His14-Glu38(α4) residues of the tetrapeptide and Aβ_16_ respectively, His protonation can affect the interaction strength. To test this, we added an extra proton to each of the three histidines in the interaction interface in 6 of the 22 Ac-HAEE-NH_2_-Aβ_16_ complex structures selected for further analysis. In the other six structures from this subset, histidine remained not charged (automatic selection of charge distribution according to force field). All 12 structures were simulated by MD for 50 ns in water, with ions (see Methods). In all systems where structures were not charged, we have observed rapid breaking of the hydrogen bonds in the Aβ_16_:Ac-HAEE-NH_2_ interaction interface, with subsequent floating of Ac-HAEE-NH_2_ to the solution. Of the charged systems, two remained stable throughout the simulation, and in the other two breakings of H-bonds between Ac-HAEE-NH_2_ and the ^11^EVHH^14^ region of Aβ_16_ occurred much later than for the systems that were not charged. In all systems where Ac-HAEE-NH_2_ drifted away from the Aβ_16_ peptide, we have observed that Ac-HAEE-NH_2_ moved back to the same ^11^EVHH^14^ interaction site, i.e., in the course of MD simulation repeated interactions occurred between them, which can be characterized as specific and transient. His14 participated in 67% (6 of 9 cases) of the repeated interactions, being more accessible than Glu11, which was mostly buried in the crease of the neighboring residues’ backbone.

Our modeling results suggest that interactions of Ac-HAEE-NH_2_ in α4β2 nAChR and of Ac-HAEE-NH_2_ with ^11^EVHH^14^ in Aβ employ similar mechanisms via identical interaction interfaces. Therefore, Ac-HAEE-NH_2_ tetrapeptide can be used as a prospective agent to modulate Aβ interaction with α4β2 nAChR.

### 2.6. Ac-HAEE-NH_2_ Tetrapeptide Prevents Aβ_42_-Induced Inhibition of α4β2 nAChR

To analyze the ability of Ac-HAEE-NH_2_ to prevent the Aβ_42_-induced inhibition of α4β2 nAChR, we used two-electrode voltage clamp in *X. laevis* frog oocytes expressing rat α4β2 nAChR. The application of 100 µM acetylcholine (ACh) to the oocytes pre-incubated with Aβ_42_ for 3 min showed an inhibition of the receptor ion current by ~30% (Figure 5A,B “Aβ_42_”). However, if the Aβ_42_ was co-applied with the 10-times molar excess of Ac-HAEE-NH_2_, the degree of inhibition was reduced significantly (Figure 5A,B “HAEE + Aβ_42_”).

More importantly, in the absence of Ac-HAEE-NH_2_, the current amplitude in the Aβ_42_-treated oocytes did not restore after a 3 min washout (Figure 5C “Aβ_42_”). However, if Ac-HAEE-NH_2_ (25–100 µM) was added to the washout buffer, the amplitude of Ach (100 µM)-evoked currents dose-dependently returned to the control levels (Figure 5A,B “HAEE after Aβ_42_”, Figure 5C “HAEE”).

At 25 µM, Ac-HAEE-NH_2_ did not affect the current amplitude, whereas at 100 µM it fully revoked the inhibition induced by Aβ_42_ (Figure 5C).

Interestingly, we found that a 3-min incubation with 10 µM Aβ_42_ increased the leakage current in *X. laevis* oocytes by 0.05–0.1 µA (Figure 5D). The increase sustained after the buffer washout of Aβ_42_, however, a consecutive washout with Ac-HAEE-NH_2_ (100 µM)-containing buffer reduced the leakage current almost to the control values. For the oocytes #1 and #2, after several incubations with Aβ_42_ and Ac-HAEE-NH_2_ washouts the overall increase in the leakage current equaled 0.05, which is consistent with the usual worn-out of the oocyte over the course of an experiment. The effect of Aβ_42_ on the membrane leakage was absent in mock-injected oocytes, thereby showing that the increase in the leakage current was because of Aβ_42_ interaction with α4β2 nAChR.

Ac-HAEE-NH_2_ and Aβ_42_ themselves did not induce any currents in α4β2 nAChR-expressing oocytes, and Ac-HAEE-NH_2_ did not affect ACh-evoked current in the absence of Aβ_42_. The observed responses in the oocytes were mediated by α4β2 nAChR, and no ACh-induced currents were detected in the mock-injected oocytes.

## 3. Discussion

Compounds that prevent interaction of Aβ with nAChRs might ameliorate the cholinergic dysfunction in AD. The development of such compounds requires the exhaustive characterization of Aβ-nAChR interaction. However, the data concerning the effects exerted by Aβ on nAChRs are contradictory, with some authors showing the activation of the receptor, while the others show the suppression of the receptor function [31]. The interaction site remains unclear, and previous findings support both the orthosteric [32] and the allosteric [7,33,34] binding to nAChRs. Molecular modelling of Aβ-nAChR interaction is also complicated due to the absence of complete or well-resolved (<3 Å) receptor structures, however, a few models of Aβ-α7 nAChR complexes were created with bioinformatics approaches [7,35,36].

It was recently found that interaction with α4β2 and α7 nAChRs is mediated by ^11^EVHH^14^ site of Aβ peptide [17,18]. The ^11^EVHH^14^ site includes three highly polar amino acid residues, of which E11 glutamate is negatively charged at physiological pH, and histidines at positions 13–14 contain a partial positive charge. Thus, we assumed that the Aβ-nAChRs interaction can be based on charge complementarity between ^11^EVHH^14^ and its counterpart motif. Charge complementarity can facilitate specific protein-protein interactions [37,38,39], stabilize a tertiary [40,41] or a quaternary [42,43] protein structure. To find charge-complementary counterparts of ^11^EVHH^14^, we screened the sequences of α4, β2 and α7 nAChR subunits. Two motifs with potential charge complementarity to ^11^EVHH^14^ were found, both in α4 nAChR subunit. Of these, ^35^HAEE^38^ motif was located extracellularly, so we hypothesized that the interaction of α4β2 nAChR with Aβ peptide can occur via (Aβ) ^11^EVHH^14^:^35^HAEE^38^ (α4) interface.

For the Aβ-α4β2 nAChR complex, no structures were proposed before, so we decided to model this interaction based on the predicted interface. For the modelling, we used the PDB:5KXI structure of α4β2 nAChR. In this structure, the ^35^HAEE^38^ site is located in an extracellular α-helix on top of the extracellular domain, and this helix remains unchanged in MD simulation. The modeling showed that ^11^EVHH^14^:^35^HAEE^38^ interface can provide a robust interaction stabilized with His-Glu H-bonds, which remained firm throughout a 100 ns MD simulation. We expected that charge complementarity would impose the parallel orientation of the motifs, but the highest stability was demonstrated by the models where ^35^HAEE^38^ and ^11^EVHH^14^ were in the anti-parallel orientation. Probably, the parallel configuration was less favorable due to the helical conformation of the ^35^HAEE^38^ site in α4β2 nAChR. ^35^HAEE^38^ site is located far from the agonist pocket, so it is hard to conclude if binding of Aβ will disrupt the attachment to the orthosteric site of the receptor. On the other hand, existing data supports the possible role of ^35^HAEE^38^ in allosteric regulation of α4β2 nAChR. N-terminal extracellular domain in α4 subunit harbors several allosteric binding sites [44,45], and a highly similar N-terminal α-helix in α7 nAChR was shown to bind negative allosteric modulators [46].

Thus, molecular modeling showed that Aβ can interact with α4β2 nAChR via ^11^EVHH^14^:^35^HAEE^38^ interface. Previously, the insights into the interaction of Aβ with α7 nAChR lead to the development of several peptide drugs aimed to prevent this binding [47,48], and we assumed that such approach could be translated to α4β2 nAChR. So, we hypothesized that Ac-HAEE-NH_2_ tetrapeptide corresponding to ^35^HAEE^38^ site in α4β2 nAChR will bind to Aβ thereby preventing its interaction with α4β2 nAChR.

Molecular docking of Ac-HAEE-NH_2_ to Aβ showed that Ac-HAEE-NH_2_ would preferentially bind to the ^11^EVHH^14^ site, confirming our assumptions based on the opposite charges of the amino acid residues in these sequences. We also detected ^4^FRHD^7^ as the potential, though the less likely binding site. ^4^FRHD^7^ amino acid composition is an anti-parallel analog of ^11^EVHH^14^, taking into account the propensity of phenylalanine to establish π-anion bonds with Glu residues [49]. In contrast with the observed anti-parallel orientation of the receptor site ^35^HAEE^38^ and the Aβ site ^11^EVHH^14^, Ac-HAEE-NH_2_ was oriented either in parallel or crosswise to ^11^EVHH^14^, suggesting that multiple binding scenarios can be realized and their exact geometry is defined by the interacting partners and their actual conformations.

Several models showed parallel orientation stabilized by three His-Glu bonds, as predicted by charge complementarity between the sequences. The MD simulation performed under physiological conditions (i.e., uncharged His residues) revealed a fast detachment of Ac-HAEE-NH_2_ from the ^11^EVHH^14^ site. However, over the 50 ns course of MD interaction, we observed that Ac-HAEE-NH_2_ goes back to ^11^EVHH^14^ and detaches again several times. This was consistent with relatively low Kd of Ac-HAEE-NH_2_ of ~10^−4^ M, as we determined using surface plasmon resonance. Such temporary, transient interaction could be nevertheless sufficient to change the functional properties of Aβ, as seen in short linear interacting motifs (SLIMs). SLIMs, or eukaryotic linear motifs (ELMs), are short protein sequences that also provide transient PPIs with Kd ranging from 10^−4^ to 10^−8^ [50]. Such motifs are crucial for recognition events such as the interaction between the members of MAP-kinase cascade, docking of src-kinase to focal adhesion kinase 1, and the pairing of transcription factors [50]. Of note, the most common length for SLIMs is 4 aa residues [51].

Though His residues can form π-anion bonds with Glu at physiological pH [52], the Aβ-Ac-HAEE-NH_2_ structures with protonated His residues demonstrated a higher stability, and half of the structural variants remained undissociated throughout the MD simulation. If such robustness is caused by salt bridges formed between positively charged His and negative Glu residues, one can assume that a peptide containing Lys or Arg at position one—the amino acids, that are positively charged at physiological pH—would bind more tightly to ^11^EVHH^14^ in Aβ. Surprisingly, RAEE peptide showed two orders of magnitude weaker binding than Ac-HAEE-NH_2_ tetrapeptide (Appendix A) and for KAEE no binding was detected at all. The forces behind such outcome are unknown, though it is possible that (1) Lys and Arg are sterically less suitable for binding and (2) in KAEE and RAEE, the unfavorable intramolecular interactions are formed between Lys/Arg and Glu [53], which disrupts a linear charge-complementary interaction. Also, pKa of His residues can shift substantially dependent on the environment. In T4 lysozyme, a His-Asp salt bridge stabilizes its tertiary structure, and the pKa of this His residue is increased to 9, meaning that it remains charged at physiological pH [54]. ^11^EVHH^14^ in Aβ is the zinc-binding center, and Ac-HAEE-NH_2_ was previously shown to prevent zinc-dependent oligomerization of Aβ, raising potential for zinc-mediated interaction between these molecules. Hence, we suggest that Ac-HAEE-NH_2_ can be used as the charge-complementary binder to ^11^EVHH^14^ and that a stable interaction can be formed under physiological conditions.

Finally, we tested the Ac-HAEE-NH_2_ effect on Aβ_42_-induced inhibition of α4β2 nAChR. For this, we used a two-electrode voltage clamp in *X. laevis* oocytes expressing α4β2 nAChR from *Rattus norvegicus*. This technique was intensively used in our previous projects to study effects on nAChRs of peptide ligands, including those produced by Aβ peptides [7,55,56]. Rat and human α4β2 nAChRs share high homology with the full conservation of “HAEE” site at the N-termini of α4 subunit. The parameters for the agonists and antagonists binding to human and rat receptors are almost identical [57,58,59], and the rat receptor has been extensively used for Aβ studies in both oocyte [60] and cellular [17] models. Thus, we consider it a relevant model for our study.

We found that 10 µM Aβ_42_ reduced the amplitude of ACh-evoked current by 30%. In comparison to the physiological levels [61], we used the relatively high (micromolar) concentration of Aβ_42_, which is consistent with the previous studies [33,60], and results from 100–5000 lower affinity of Aβ_42_ to α4β2 nAChR than to α7 nAChR [32]. As shown before [33], the inhibition of α4β2 nAChR by Aβ was partially reversible, meaning that a single 3-min washout was not sufficient to restore the current amplitude. Both the amplitude of α4β2 nAChR-mediated current, and the degree of the receptor inhibition by Aβ_42_ are in agreement with the previously published results [60].

We found that co-administration of Aβ_42_ with 10-times molar excess of Ac-HAEE-NH_2_ reduced the inhibitory effect of Aβ_42_ by half, thereby confirming the ability of Ac-HAEE-NH_2_ to prevent α4β2 nAChR inhibition by Aβ. Compared to YEVHHQ peptide that mimics the other side of α4β2 nAChR–Aβ interface [17], Ac-HAEE-NH_2_ did not induce any currents itself, which can be beneficial to avoid the potential side effects. If co-applied with Aβ_42_, Ac-HAEE-NH_2_ did not fully repair the receptor function, which is possibly due to its relatively low affinity to Aβ_42_. However, the washout of Aβ_42_ with Ac-HAEE-NH_2_-containing buffer completely restored the receptor response. The Ac-HAEE-NH_2_ washout was most effective at 100 µM of the peptide, and less so at 50 µM, thus, a high molar excess of Ac-HAEE-NH_2_ over Aβ_42_ is required to exert its effect. The concentration of soluble Aβ species in the brain ranges from pM to nM [62,63], and peptide drugs are well-tolerated in hundreds of micromoles per liter, so the required concentration of Ac-HAEE-NH_2_ in the brain can probably be reached without adverse effects.

Aside from lowering the response amplitude, Aβ_42_ induced the increase in leakage current in the oocytes. It was previously shown that Aβ_42_ can interact with lipid membranes and form ionic-permeable channels [64,65], which could have explained the observed effect. However, Aβ_42_ did not alter the leakage current in untransfected oocytes, thus suggesting the leak was due to the interaction of Aβ_42_ with α4β2 nAChR. Apparently, Aβ_42_ disrupts the proper gating of α4β2 nAChR, as it was previously shown for ryanodine receptor-dependent calcium leaks in the endoplasmic reticulum [66]. The washout with Ac-HAEE-NH_2_ peptide restored the Aβ-induced receptor leak, which is more evidence for Aβ_42_-α4β2 nAChR complex disruption by Ac-HAEE-NH_2_. Previously, we observed that injections of Ac-HAEE-NH_2_ effectively reduce the amyloid load in the brains of AD model mice [16]. The formation of Aβ-α4β2 nAChR complexes might be connected with amyloid formation, with such complexes either serving as aggregation seeds or promoting neuronal death [67,68] Thus, considering the results of the current study, the anti-amyloid effects of Ac-HAEE-NH_2_ could be linked to its ability to prevent the interaction of Aβ with α4β2 nAChR.

Interactions of soluble Aβ species with target proteins bear a pathological significance in Alzheimer’s disease [69,70,71], and targeting these interactions represents a promising therapeutic strategy [69,72,73,74]. The data obtained in the current study suggests that Aβ-α4β2 nAChR interaction is mediated by the charge complementary interface (Aβ) ^11^EVHH^14^:^35^HAEE^38^ (α4). Tetrapeptide Ac-HAEE-NH_2_, which is the synthetic analog of the receptor side of this interface, proved to efficiently repair the Aβ-dependent loss of cholinergic function in α4β2 nAChR-transfected oocytes. The findings of the study provide a prospective drug candidate for treatment of cholinergic deficit in AD (Figure 6).

## 4. Materials and Methods

### 4.1. Preparation of Aβ Peptides

Synthetic Peptides Aβ_16-G4-C_ [Ac]-DAEFRHDSGYEVHHQKGGGGC-[NH2] and Aβ_42_[H2N]-DAEFRHDSGYEVHHQKLVFFAEDVGSNKGAIIGLMVGGVVIA-[COOH] were obtained from Biopeptide (San Diego, CA, USA). For the electrophysiology experiments, Aβ_42_ peptide was monomerized as described previously [7]. A fresh 5 mM solution of Aβ_42_ was prepared by adding 10 µL of 100% anhydrous dimethyl sulfoxide (DMSO) (MilliporeSigma, St. Louis, MO, USA) to 0.224 mg of the peptide, followed by incubation for 1 h at room temperature to completely dissolve the peptide. For use in a direct binding assay, lyophilized Aβ_16-G4-C_ was dissolved in 10 mM sodium acetate buffer, pH 4.5, to reach a concentration of 0.05 mg/mL.

### 4.2. Ac-HAEE-NH_2_ and Other Tetrapeptides

Ac-HAEE-NH_2_ and other tetrapeptides (Appendix A) with charge complementarity to ^11^EVHH^14^ region of Aβ were obtained from Verta Ltd. (St. Petersburg, Russia). All tetrapeptides were stabilized by N-terminal acetylation and C-terminal amidation, thus referred to as Ac-XXXX-NH_2_ (Ac-HAEE-NH_2_). To prepare stock solutions for the surface plasmon resonance experiments, the lyophilized tetrapeptides were dissolved in sterile water to reach a concentration of 10 mM, filtered through a 0.22 µM filter (MilliporeSigma, St. Louis, MO, USA) and stored in a freezer at −80 °C.

### 4.3. nAChR Protein Sequence Analysis

The screening of α4β2 nAChR and α7 nAChR sequences for the motifs with charge complementarity to ^11^EVHH^14^ in Aβ was performed with a ScanProsite tool (https://prosite.expasy.org/scanprosite/) using [HRK]-[VALI]-[DE]-[DE] as a query on FASTA-formatted protein sequences of α4, β2 and α7 nAChR subunits of *Homo sapiens*, obtained from UniProt (https://www.uniprot.org/).

### 4.4. Bioinformatics

#### 4.4.1. Structure Modelling

The structure of α4β2 nAChR neuronal acetylcholine receptor was modelled using as a template PDB:5KXI structure [75] solved by X-ray crystallography with a resolution of 3.941 Å. Fragments 1–24 and 365–585 of the α4 subunit, and 1–25 and 356–445 of the β2 subunit are absent in this structure. The missing fragments were modeled by the SwissModel, RaptorX, and iTasser servers, in accordance with our previously developed approach [76], which involves the construction of models by several independent servers with subsequent analysis of the quality of structures and identification of a representative model. Using this model, expert modeling of the final structure and energy minimization in the Amber12 force field was performed. The extracellular domain was isolated from the full protein model and its structure was equilibrated by molecular dynamics (MD).

The initial Ac-HAEE-NH_2_ tetrapeptide was obtained from the α4β2 nAChR model structure. Hydrogens, acetyl and amino (CH_3_CO and NH_2_ respectively) end groups were added and the resulting structure minimized in water with the AMBER99SB-ILDN force field. Then it was processed in the production run of MD for 100 ns using the Gromacs package.

Two models of Aβ_16_ were taken from PDB:1ZE7 solution NMR structure (models 1 and 7), and hydrogens, acetyl and amino end groups were added. The difference in the structures of these two models is in the position of N-terminus. Model 1 represents a folded, circular-shaped conformation with its N-terminus close to the C-terminus, and the model 7 structure is more unfolded with its N-and C-termini further away from each other. These structures were subsequently used as receptors for the docking of Ac-HAEE-NH_2_ tetrapeptide.

The previously created model of Aβ_42_ [7] was further equilibrated by molecular dynamics. Structures used as templates for the initial expert modelling of Aβ_42_ were selected from the data of our analysis of Aβ structures in the PDB [77].

#### 4.4.2. Interactions Modelling

##### Aβ_42_—α4β2nAChR Interaction Modelling

Modelling the interaction interface centered on the Aβ_42_
^11^EVHH^14^ and α4β2 nAChR ^35^HAEE^38^ interaction sites was performed according to the following protocol.

(1) Targeted global docking of Aβ_42_ with α4β2 nAChR using PatchDock [78] and HADDOCK [79] servers. From the dataset of modelled structures of the complex, a subset of structures was selected where several H-bonds were formed between ^10^YEVHHQ^15^ and ^35^HAEE^38^ primarily via the histidine—glutamic acid residues. (2) Refinement of the resulting structures of Aβ_42_ with Rosetta server [80] and relaxing them during 20 ns of MD production run using the Gromacs package. (3) Local docking of the ^10^YEVHHQ^15^ fragment from Aβ_42_ to α4β2 nAChR extracellular domain using AutoDock Vina 1.1.2 [81]. (4) The structure of Aβ_42_ from (2) was superposed with the YEVHHQ fragment from (3), so as to achieve superposition of the backbone atoms of residues TYR10 and GLN15 of the ^10^YEVHHQ^15^ segment in Aβ_42_ and the YEVHHQ-α4β2 nAChR docked structure. The ^11^EVHH^14^ segment in Aβ_42_ was substituted with EVHH of the α4β2 nAChR-YEVHHQ structure. Several consecutive energy minimization steps on single residues were run to optimize the conformation of the HAEE-EVHH interface. All clashes between α4β2 nAChR and Aβ_42_ were removed by rotating the N-terminal (1–9) and C-terminal (16–42) parts of the Aβ_42_ structure, and then energy minimization was run for the full system. (5) Local docking with Rosetta server was performed on the resulting Aβ_42_-α4β2 nAChR model to fine-tune the conformation of the N- and C-terminal parts of Aβ_42_. (6) The final structures were simulated for 100 ns of MD production run using the Gromacs package and the AMBER99SB-ILDN force field.

##### Ac-HAEE-NH_2_ Docking to Aβ_16_ and Aβ_42_

Energy minimized and relaxed Ac-HAEE-NH_2_ tetrapeptide was docked to Aβ_16_ and Aβ_42_ structures with several docking servers and programs running global and local docking. Acetyl and amino groups from the N- and C-termini of the receptor and ligand were removed when such input requirements were specified for some of the docking servers. Full blind global docking of Ac-HAEE-NH_2_ to Aβ_16_ and Aβ_42_ was performed with PatchDock, ClusPro [82], GrammX [83], SwarmDock [84] servers, and HEX package [85]. Global docking with the target docking site specification was run using ClusPro, SwarmDock, HADDOCK and PatchDock servers. Local docking was performed with AutoDock Vina 1.1.2 (Scripps Research, San Diego, CA, USA). For Aβ_16_ docking was done twice for the two models (1 and 7) from the NMR dataset of PDB:1ze7. The docking results were processed and analyzed using the in-house server QASDOM [20].

#### 4.4.3. Molecular Dynamics

All structures taken for molecular dynamics simulations were energy minimized consecutively with the steepest descent and conjugated gradients algorithms and equilibrated in water with the NaCl concentration of 115 mM under position restraints for 1 ns in the constant volume (NVT) and the constant pressure (NPT) ensembles respectively. The AMBER99SB-ILDN force field was used for all runs. Simulations were carried out using the particle mesh Ewald technique with repeating boundary conditions and 1 nm cut-offs, using the LINCS constraint algorithm with a 2-fs time step. Two coupling and energy groups were used, a constant temperature of 300 K was maintained. All computations were performed using the Gromacs package (University of Groningen, Groningen, The Netherlands).

### 4.5. Direct Binding Assay

Surface plasmon resonance (SPR) was utilized to detect the direct binding of the tetrapeptides to immobilized Aβ_16-G4-C_. All SPR experiments were carried out on a BIAcore T100 instrument (GE Healthcare, IL, USA). Research grade sensor chips CM5 carrying the hydrophilic carboxymethylated dextran matrix, HEPES Buffered Saline (HBS) buffer (10 mM HEPES (4-(2-hydroxyethyl)-1-piperazineethanesulfonic acid)), pH 7.4, 150 mM NaCl, 3 mM EDTA, 0.005% surfactant P20), 1-ethyl-3-(3-dimethylaminopropyl)-carbodiimide (EDC), N-hydroxysuccinimide (NHS), 2-(2-pyridinyldithio)-ethaneamine (PDEA), and cysteine were purchased from Biacore (GE, Boston, MA, USA). All other chemicals and solvents were of HPLC-grade or better and were obtained from MilliporeSigma (St. Louis, MO, USA). All buffers were filtered (0.45 μm, nylon) prior to use.

Attachment of the synthetic peptide Aβ_16-G4-C_ to the CM5 chip was performed according to the thiol bond formation protocol described in the Sensor surface handbook (GE Healthcare, Chicago, IL, USA). The carboxymethyl dextran matrix was activated by injection of a 1:1 mixture of EDC and NHS (30 μL, 400 mM EDC, 100 mM NHS) with the following injection of an 80 mM PDEA solution in 0.1 M sodium borate (pH 8.5). The Aβ_16-G4-C_ solution was then injected into the activated flow cell (0.05 mg/mL peptide in 10 mM sodium acetate buffer, pH 4.5). Unreacted disulfide groups on the CM5 chip surface were capped with a 50 mM cysteine solution in 0.1 M sodium acetate buffer (pH 4.0). The change corresponding to the immobilization of Aβ_16-G4-C_ was 1023 response units (RU). The flow rate used for all immobilization steps was 5 μL/min. An unmodified dextran surface was used as a reference surface.

Then, the binding affinities of the immobilized Aβ_16-G4-C_ to the following peptide-analytes were measured. Samples of Ac-HAEE-NH_2_ and other tetrapeptides were prepared by dilution of respective stock solutions in the running buffer (10 mM HEPES, pH 6.8). Each analyte was diluted to different concentrations (0 μM, 50 μM, 100 μM, 200 μM, 500 μM, 1000 μM, 1500 μM, 2000 μM) and injected in multichannel mode (volume 50 μL and rate 10 μL/min) during 300 s. Then, the chip surface was exposed to the running buffer without analyte for 120 s. After each injection of the analyte, the surface was regenerated with 5 μL of the regeneration buffer (HBS buffer containing 10 mM HEPES, 3 mM EDTA, 0.005% surfactant P20 and 150 mM NaCl, pH 7.4). The signal from the reference surface was subtracted from the raw data, obtained from the flow cell with the immobilized ligand.

The kinetic rate constants were calculated from the sensorgrams by globally fitting the response curves obtained at various analyte concentrations using the Langmuir model (1:1 binding) in the BIAevaluation 4.1 program. The association (k_on_) and the dissociation (k_off_) rate constants were fitted simultaneously (1),
dR/dt = k_on_ C (Rmax − R) − k_off_ R(1)
where R stands for the biosensor response of the formed complex, C is the concentration of the analyte, and Rmax is the maximal theoretical value of the binding response for a given analyte.

Using the obtained data dissociation (Kd) constant was calculated from the ratios of the association (k_on_) and dissociation (k_off_) rate constants: Kd = k_off_/k_on_, Ka = k_on_/k_off_.

### 4.6. Electrophysiology

Two-electrode voltage clamp electrophysiology on the α4β2 nAChR expressed in *Xenopus laevis* oocytes was performed according to previously published protocols [7]. Stage V ± VI *Xenopus laevis* oocytes were defolliculated with 1 mg/mL collagenase Type I (Life Technologies, Carlsbad, CA, USA) at room temperature (21–24 °C) for 2 h in Barth’s solution without calcium (88.0 mM NaCl, 1.1 mM KCl, 2.4 mM NaHCO_3_, 0.8 mM MgSO_4_, 15.0 mM HEPES/NaOH, pH 7.6). The oocytes were stored in Barth’s solution with calcium for 72–120 h (88.0 mM NaCl, 1.1 mM KCl, 2.4 mM NaHCO_3_, 0.3 mM Ca(NO_3_)_2_, 0.4 mM CaCl2, 0.8 mM MgSO_4_, 15.0 mM HEPES/NaOH, pH 7.6) supplemented with 63.0 μg/mL penicillin-G sodium salt, 40.0 μg/mL streptomycin sulfate.

Oocytes were injected with 3 ng plasmids coding the rat α4 and β2 nAChR subunits (pcDNA3.1 vector) in a molar ratio of 1:1 using an Auto-Nanoliter Injector NanoJect-2 (Drummond Scientific Company, Broomall, PA, USA) in a total injection volume of 23 nL. After injection, oocytes were incubated at 18 °C in Barth’s solution with calcium for 48–120 h. Electrophysiological recordings were made using a Turbo TEC-03X amplifier (npi electronic GmbH, Tamm, Germany) and WinWCP recording software (University of Strathclyde, Glasgow, UK). Oocytes were placed in a small recording chamber with a working volume of 50 μL and 100 μL of agonist (acetylcholine) solution in Barth’s buffer were applied to an oocyte. Oocytes were pre-incubated with Aβ_42_ (10 µM) or Ac-HAEE-NH_2_ (25, 50 or 100 µM) for 3 min followed by its co-application with acetylcholine (100 µM). To allow receptor recovery from desensitization, the oocytes were superfused for 5–10 min with buffer (1 mL/min) between ligand applications. Electrophysiological recordings were performed at a holding potential of −60 mV.

### 4.7. Statistical Analysis

Data are presented as means of at least three independent experiments ± SD. The comparison of data groups in electrophysiology studies was performed with ordinary one-way ANOVA. Post-hoc analysis was performed with the Tukey test. Shapiro-Wilk test was used to confirm the normality of the dataset. Statistical analysis was performed using GraphPad Prism 8.4.1 software (GraphPad Software Inc., CA, USA).

## Figures and Tables

**Figure 1 ijms-21-06272-f001:**
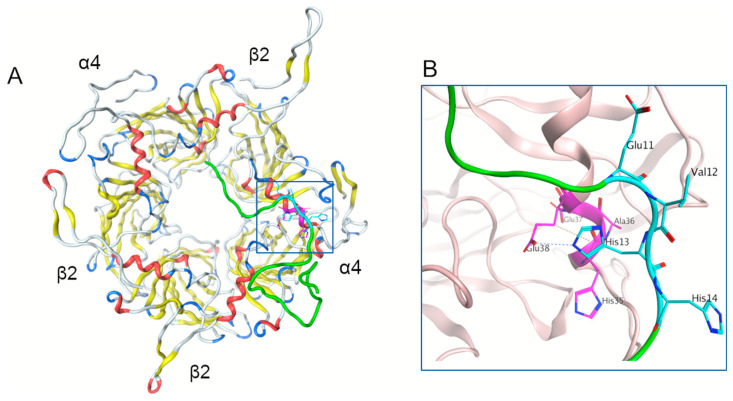
Model of the interaction of the α4β2 nAChR site ^35^HAEE^38^ with Aβ_42_ after 100 ns of molecular dynamics structure equilibration. (**A**) Model of the α4β2 structure with bound Aβ_42_ peptide, viewed from the extracellular side. (**B**) Detailed view of the interaction interface. The Aβ_42_ peptide is colored green with the ^11^EVHH^14^ site shown in cyan. The ^35^HAEE^38^ site is colored magenta. The N-terminal α-helix of both α4 and β2 subunits is colored red.

**Figure 2 ijms-21-06272-f002:**
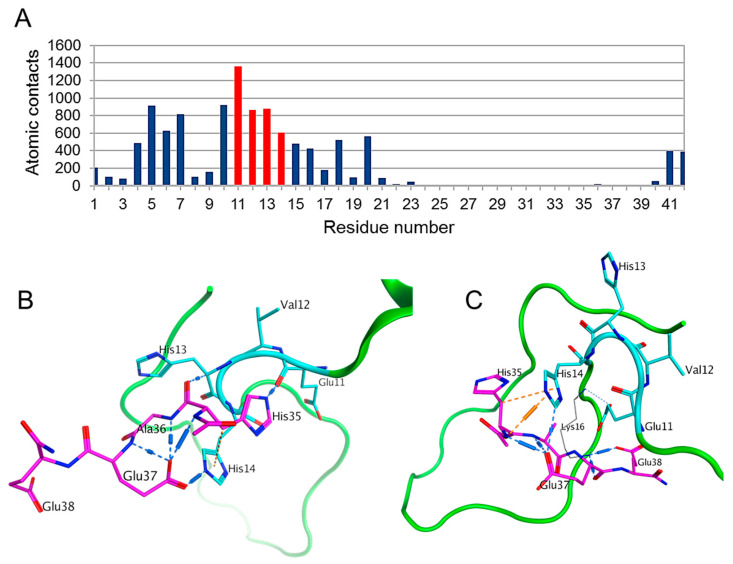
Global docking of Ac-HAEE-NH_2_ to Aβ_42_. (**A**) A histogram of Aβ_42_ atomic contacts to the Ac-HAEE-NH_2_ tetrapeptide for the data from six docking servers. The position of the ^11^EVHH^14^ site is highlighted in red. Calculated by QASDOM [20] metaserver. (**B**,**C**) Examples of the docked Ac-HAEE-NH_2_ peptide. The Aβ_42_ peptide is colored green, with the ^11^EVHH^14^ site shown in cyan, and the Ac-HAEE-NH_2_ tetrapeptide is colored magenta.

**Figure 3 ijms-21-06272-f003:**
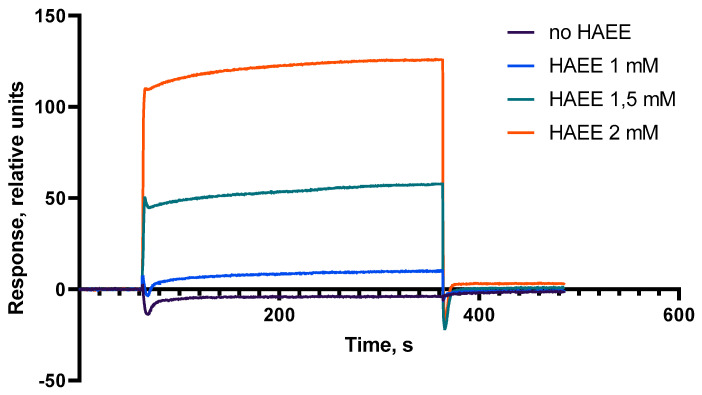
Sensorgrams showing direct binding of Ac-HAEE-NH_2_ (1 mM–2 mM) to immobilized Aβ_16_. Spikes at the start and end of Ac-HAEE-NH_2_ injections are due to a slight time delay in the reference cell and appear when reference subtraction is carried out.

**Figure 4 ijms-21-06272-f004:**
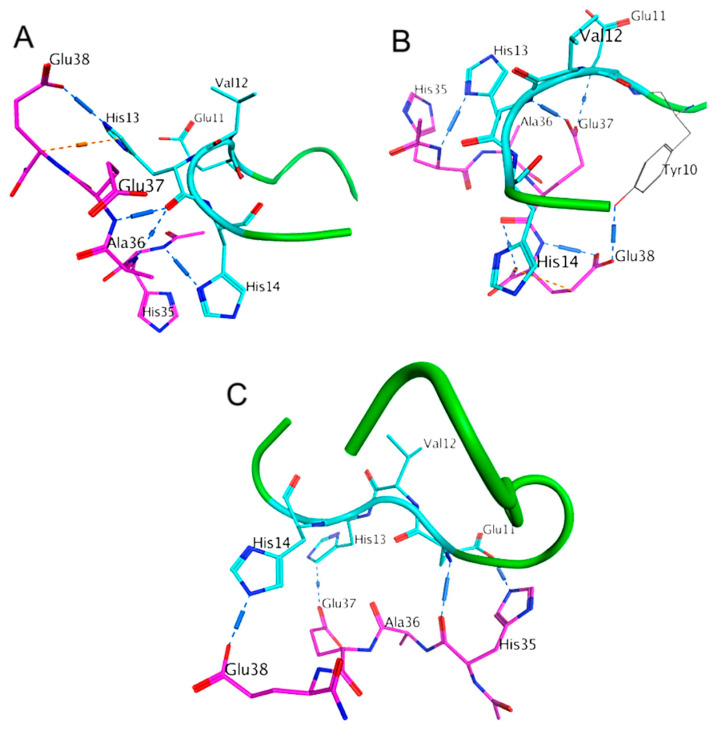
Global docking of Ac-HAEE-NH_2_ to Aβ_16_ (**A**,**B**) Examples of the docked Ac-HAEE-NH_2_ peptide. The Aβ_16_ peptide is colored green with the ^11^EVHH^14^ site shown in cyan, and the Ac-HAEE-NH_2_ tetrapeptide is colored magenta. (**C**) The proposed interface of HAEE-EVHH interaction based on a docking model.

**Figure 5 ijms-21-06272-f005:**
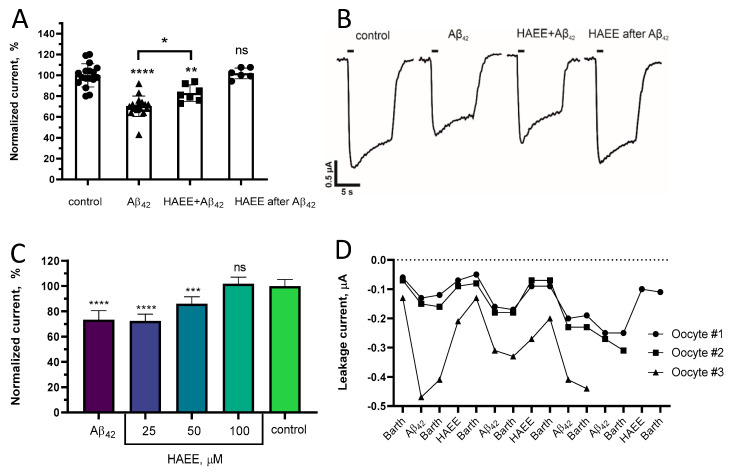
(**A**) Representative ion current traces and (**B**) normalized amplitudes of ACh (100 μM)-induced ion currents in α4β2 nAChR-expressing *Xenopus laevis* oocytes in control and after 3 min pre-incubation with 10 µM Aβ_42_ (“Aβ_42_”), 10 µM Aβ_42_ and 100 µM Ac-HAEE-NH_2_ (“HAEE + Aβ_42_”), or 10 µM Aβ_42_ followed by washout with Barth’s solution containing 100 µM of Ac-HAEE-NH_2_ (“HAEE after Aβ_42_”). (**B**) Individual current amplitude values are depicted as black dots. (**C**) Normalized ACh (100 μM)-induced current amplitudes in α4β2 nAChR-expressing *Xenopus laevis* oocytes in control and after 3 min pre-incubation with 10 µM Aβ_42_, followed by 3 min washout with Barths’ solution in the absence (“Aβ_42_”) or presence (“HAEE”) of Ac-HAEE-NH_2_. (**A**,**C**) Data are presented as mean ± SD, n ≥ 3. *—*p* < 0.05, **—*p* < 0.005, ***—*p* < 0.001, ****—*p* < 0.0001, ns—nonsignificant. (**D**) The leakage current in α4β2 nAChR-expressing *Xenopus laevis* oocytes was measured after 3 min consecutive incubations in Barth’s solution (“Barth”), in Barth’s solution containing 10 µM Aβ_42_ (“Aβ_42_”), and in Barth’s solution in the absence (“Barth”) and presence of 100 µM Ac-HAEE-NH_2_ (“HAEE”).

**Figure 6 ijms-21-06272-f006:**
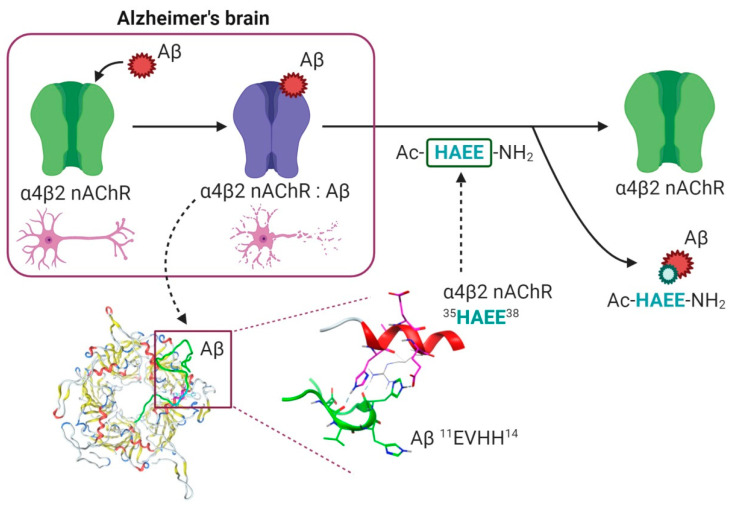
The possible role of ^11^EVHH^14^:^35^HAEE^38^ interface in cholinergic deficit associated with Alzheimer’s disease. In brains of Alzheimer’s disease patients, interaction of Aβ with nAChRs causes transition of the receptor from functional state (green) to dysfunctional state (violet), which may lead to selective loss of cholinergic neurons (top left). Our results suggest that the interaction of Aβ with α4β2 nAChR is mediated by charge complementary interface ^11^EVHH^14^:^35^HAEE^38^ (bottom left and middle) and that Ac-HAEE-NH_2_ peptide corresponding to this interface can competitively displace Aβ from the complex and restore the functionality of α4β2 nAChR (top right).

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
