# Peer review of "Tetrapeptide Ac-HAEE-NH2 Protects α4β2 nAChR from Inhibition by Aβ"

_ijms, 2020, doi:10.3390/ijms21176272_

Round 1

Reviewer 1 Report

Previous studies have demonstrated an overlap in localisation of the nACh receptors of the α4β2 subtype and aggregated forms of Aβ peptides in the brain of Alzheimer’s disease patients. Moreover, there are experimental evidence that human Aβ via its EVHH site can interact with this receptor, and a synthetic Aβ fragment containing EVHH sequence is neuroprotective. The latter is most probably due to an ability of this fragment to interfere with nAChRα4β2/Aβ interaction. Barykin and colleagues carried out bioinformatic analysis of potential EVHH interaction sites within the nAChRα4β2 sequence followed by testing in vitro molecular interaction between the best hit and Aβ16 fragment. In the submitted manuscript authors provide evidence that that the HAEE site of nAChRα4β2 is the interaction site of human Aβ42. Importantly, a synthetic HAEE tetrapeptide was able to block interaction of the extracellular domain of nAChRα4β2 expressed in frog oocytes with externally added Aβ42, preventing potentially detrimental inhibition of the acetylcholine-induced receptor ion current. Authors suggest that HAEE peptide or its analogues might be used for restoration of the cholinergic dysfunction typical for Alzheimer’s disease. Overall, the study is well executed and documented, and the paper will be of interest researchers in the field.

My main concern is the discrepancy between using in the bioinformatic analysis of structural data for HUMAN nAChRα4β2 and RAT receptor for experiments in oocytes. This sounds odd and should be explained.

Minor

115 – “… primarily through His-Glu interaction …”  -  which particular there are two histidine residues in the Aβ interaction site and two glutamic acid residues in the receptor interaction site. Which of these residues and in what combination are actually involved in the interaction between these sites?

127  - “ … H-bonds formed by histidine …” -  Similarly to the above, which histidine? Or all of them?

136 – “… N-acetylated and C-amidated for increased stability …” –  it is not clear what kind of stability authors are talking about. If it is about stability in vivo, why this was important for short-term oocyte experiments used by authors? Is non-modified peptide extremely unstable?

147 – “… participated in the interactions (Fig. S1, C and D)” – I can see only single panel A in the Suppl Figure S1!

159 – “…We have shown earlier that fragment 1-16 of Aβ (Aβ16) represents an adequate model for in vitro …” – more information about this peptide and its use in both in vitro and in vivo models should be provided here.

199 – “… Since the modelling results showed the presence of H-bonds between His and Glu residues …” – once again, residues should be explicitly specified. Both histidine and glutamic acid residues are present in the receptor site, which makes the current statement ambiguous (for example, it can be interpreted as H-bonds between residues from the binding sites of two receptor molecules).

205 – “… we have observed rapid breaking of the hydrogen bonds …” – I guess that these are hydrogen bonds at the interaction interface between two molecules – this should be specified.

303 – “In contrast with the anti-parallel orientation of 35HAEE38 and 11EVHH14, Ac-HAEE-NH2 was oriented …” -  this is very confusing statement because it is unclear where authors talking about the synthetic peptide and where – about a stretch of amino acids in the actual receptor molecule.

373 – “Interactions of soluble Aβ species with target proteins bear a pathological significance in Alzheimer’s disease, andtargeting these interactions represents a promising therapeutic strategy”- such strong statements should be only if they can be properly supported by the existing literature. Or avoided altogether.

Reviewer 2 Report

In this paper, the authors found that 35HAEE38 is a potential binding site for Aβ on α4β2 nAChR and Ac-HAEE-NH2 tetrapeptide corresponding to this site is a potential therapeutic for the treatment of α4β2 nAChR-dependent cholinergic dysfunction in AD.

The paper is well written and interesting to read, however, there are several questions needed to be addressed before publishing:

  1. In Figure 1A, the author showed the docking structure of the α4β2 with bound Aβ42 peptide. As far as I know, multiple structures will be generated by docking. What is the probability (percentage) of the structure showed in Figure 1A?
  2. In section 2.4, the Kd was investigated by using Aβ16 instead of Aβ What is the reason using Aβ16? Are there any technical problems of using Aβ42?
  3. In this paper, the author provided the parameters as kon= 0.37, koff=0.04x10-3. What is the units for both kon and koff? Even, we consider the units of koff as s-1, the lifetime of the complex is 25*10^3 s(1/koff). This means the system has strong mechanical attachment. So, this is not consistent with the author’s simulation results “However, over the 50 ns course of MD interaction, we observed that Ac-HAEE-NH2 goes back to 11EVHH14 311 and detaches again several times”.

Round 2

Reviewer 2 Report

The authors answer the comments very precisely. I recommend this paper published in IJMS.